# Gross domestic product and logistics performance index drive the world trade: A study based on all continents

Ruwan Jayathilaka[1]*, Chanuka Jayawardhana[2], Nilupul Embogama[2], Shalini Jayasooriya[2], Navodika Karunarathna[2], Thisara Gamage[2], Nethmali Kuruppu[2]

**1** Department of Information Management, SLIIT Business School, Sri Lanka Institute of Information Technology, Malabe, Sri Lanka, **2** SLIIT Business School, Sri Lanka Institute of Information Technology, Malabe, Sri Lanka

\* ruwan.j@sliit.lk

**Data Availability Statement:** All relevant data are within the paper and its Supporting Information files.

## Abstract

The purpose of this study was to examine the impact of Gross Domestic Product (GDP) and Logistics Performance Index (LPI) on international trade of nations across each continent and worldwide. Secondary data collected on 142 countries—37 Asian, 41 European, 41 African, 3 Oceania, 14 Middle East, 11 North American and 9 South American–were analysed across the years 2007, 2010, 2012, 2014, 2016, and 2018. Panel regression technique was applied and the random effect (RE) model was chosen based on the results of the Hausman tests and Breusch–Pagan Lagrange Multiplier test. The findings revealed that the LPI has a positive relationship with net exports globally and specifically within the continents of Asia, Europe, and Oceania. Moreover, while the GDP appears to have a significant negative impact on net exports, specifically within Asia, in contrast, countries in Oceania and the Middle East present a positive relationship. Also on the African continent, GDP has a significant negative impact on the net exports. Findings provide a holistic picture of the impact of LPI & GDP on net exports, which will assist governments in the formulation and revision of its strategies and policies to expedite the growth of exports and in turn, the economy. This study was the first of its kind to explore the impact of GDP and LPI on international trade of nations across worldwide.

## Introduction

International trade plays a pivotal role in all economies. Without it, the world would come to a standstill and such importance has paved way for a multitude study in the area and its facilitation towards global economic conditions. Present day economists regard logistics as one of the most crucial aspects in the growth of international trade. Logistics is the strategic management of the transportation and storage of resources, components, completed inventory, and related information flows across companies and marketing channels, thus enabling countries to trade industrial, agricultural and various other consumer products in global markets. Logistics Performance Index (LPI)–a benchmarking tool to identify the performance of trade logistics

**Funding:** The authors received no specific funding for this work.

**Competing interests:** The authors have declared that no competing interests exist.

within a nation, Gross Domestic Product (GDP)–market value of all goods produced within a territory and the Purchasing Power Parity (PPP)–metric to analyze the differences in the currencies of nations, all act as acceptable measures for identifying the emerging trends and the status of an economy. Insights on these indicators assists to enhance the quality of decisions taken by authorities to improve the international trade of a nation.

The volume of international trade is largely reliant on the variables that facilitate trade and lower trade costs in relation to transportation, communication, exchange rates, regulations, etc. [1]. While highly efficient logistics services reduce trade costs, promote product movement, ensure product safety and delivery speed [2], facilitation of cross-border trade is aided by efficient customs administration and regulatory authorities, telecommunications, infrastructure quality, and competent logistics Felipe and Kumar [3]. Moreover, when imports and exports move in and out of borders multiple times in the form of intermediate and final goods, facilitating trade assist in maintaining trade development costs at a minimal.

The purpose of this study is to analyze the impact of GDP and LPIs on international trade across continents relative to global countries. The LPI analyses the differences between countries, providing a general picture of customs procedures, logistics costs, and the quality of the infrastructure necessary for overland and maritime transport [4]. It also provides information to clearly understand the global logistic performance trends over time. LPI also includes a set of domestic performance indicators that help to measure the domestic logistic trend periodically [5]. The GDP provides an economic snapshot of a country, that can be used to estimate the size of an economy as well as the growth rate and the direction of economic growth. The GDP can be calculated using three perspectives: based on expenditure, production, or income and can be adjusted for inflation and population to provide deeper insights into economic performance. Despite its limitations, GDP is a key tool to guide policymakers, investors, and businesses in strategic decision-making. The Purchasing Power Parity (PPP) associates currencies of various nations using a "basket of goods" enabling economists to cross-compare the economic output and living standards across nations. Several nations alter their GDP numbers to reflect the PPP through the conversion of nominal GDP into a number which allows to make real comparisons among the countries with different currencies and provides a more realistic depiction of a country's overall well-being.

Recently much research has focused continent-wide relationship between the LPI and international trade. Yet, many continents lack empirical studies in this area of research and findings of the LPI, GDP, and PPP when their impacts are considered together. This dearth of literature prompted the authors to conduct this research. Thus, the study findings contribute to bridging the empirical gaps in this subject area.

This article is authored according to the following structure: section 1; the introduction, contained details about the basic background and the outline of the study; section 2 presents an overview of the relevant literature; Section 3 discusses the methodology, where the LPI, GDP, and PPP are deliberated with Net Exports (NEX) globally and within each continent; Section 4 presents the results and a discussion of the outcomes that describe the relationship between the LPI, GDP, PPP and the NEX; and Section 5 is the conclusion which provides the summary and implications of the research.

## Literature review

The literature pertaining to international trade flows, repeatedly stresses that the logistics performance, GDP and the PPP have an impact on the volume of international trade. Logistics has been featured as a critical factor in the facilitation of trade and in turn, a stimulator of a nation's economic development. Thus, it significantly influences bilateral trade flows [1,6,7].

The LPI is a comprehensive index designed to assist countries in identifying challenges and opportunities in trade logistics work evidence [8–11]. Further, previous studies highlight that a rise in the GDP of the trading partners escalates the export trade volumes [12,13]. However, the literature indicates a unidirectional causality from export to import, meaning that in the long run, export leads to import but not vice versa [14]. Furthermore, exchange rates are at the core of scholarly work related to international trade, and exchange rates are said to adjust at a level set as per PPP [15]. A discussion of existing literature pertinent to the variables studied in this research are discussed continent wise henceforth.

## Asia-pacific

Asia being a significant contributor to the global economy makes a substantial impact in perpetuating competitive prices within the global export market [16]. Studies indicate that progress in the exporting country's LPI performance can explain the rise in bilateral trade in Central Asia, as well as the export basket sophistication [3,17]. On the contrary, studies indicate that the LPI does not make a significant impact on trade [2]. Developments in logistics services deepen cross-border International Production Networks (IPNs) and in turn, boost trade volumes. However, a focal finding in the literature is the need for further infrastructure development associated with logistics within the Asian region to expedite the growth of exports [18,19]. Further, landlocked countries–Central Asian economies and a few South and Southeast Asian economies require to pass through another country(transit) to access major international transport lanes and connect with the global markets. Thus, these countries are reliant on the infrastructure availability of transit countries faces additional challenges in cross-border trade [20].

A study on ASEAN Free Trade Area (AFTA) member countries further uncovers that the population, GDP and the language too, elucidate export movements [21]. Another study across ASEAN countries revealed that custom-related barriers such as time-consuming documentation requirements, burdensome inspection requirements & inefficient inbound clearance processes as well as mode-specific barriers such as Aviation cabotage regulations and limitations imposed on fleet size, equipment usage and hours of operation hinder the trade relations [22]. Other studies have explored the influence of corruption on trade facilitation and revealed that corruption significantly affects the LPI [23].

When focusing on Oceania and Pacific, it can be observed that the long-term economic successes of several Pacific Island countries are weak, and some countries have been experiencing slow-paced growth for years. Investigating on the overall logistical performance, Dimitrievska, Mihajlović [24] has found that Oceania countries (together with Asian countries) come in second to the Europe region. Imports from Australia to Pacific Islands are heavily influenced by their population and per capita GDP [25]. The findings of the study imply that the population of Pacific Island countries and Australia, as well as the infrastructure of Pacific Island countries and their distance from Australia, have a substantial impact on their exports.

## EU

The significant impact that LPIs of trading partners has on the volume of bilateral trade within the European Region has been established [1]. Bensassi, Márquez Ramos [26] highlighted that logistics measures are important at the regional level and the number, size and quality of logistics facilities have a beneficial positive impact on export flows. Therefore, it is critical to invest in the development of railway infrastructure in the Central and Eastern European Member States to assure long-term transport efficiency, establish competitive advantages, and consequently stimulate long-term economic growth [27]. Saidi, Mani [28] too reveals that increases

in transport infrastructure results in enhanced economic growth. Central and Eastern Europe countries possess strong economic relationships with countries in Western Europe resulting in increased employment and technology transfers. This can be further enhanced by developing infrastructure logistics which in turn strengthens connectivity across the regions. Other studies have shown that the LPI sub-components of logistics quality and competence as well as international shipment have positive significant impact on trade volumes [1].

Marti, Puertas [4] observing the relationship between LPI and export competitiveness in the EU-26 claimed that logistics are more significant for exporting countries than for importing countries. These findings align with the study by Celebi [29] conducted across the low–income and high-income countries. Moreover, studies within the European region has underlined that the GDP and population levels are key contributors to export flows [30].

## Middle east

Over the past three decades, the United Arab Emirates (UAE) has enjoyed strong economic growth and significant export expansion. By 2012, the GDP of the UAE grew 25 times compared to its level in 1975 [31]. Bi-directional causalities were observed between GDP and exports and imports of Middle Eastern countries in the long term for Cyprus, Egypt, Iran, Israel, Jordan, Oman, Qatar, Saudi Arabia, Turkey, and Yemen, but not for Bahrain, Kuwait, and Lebanon [32]. With the use of gravity models and the LPI index as a good proxy for trade facilitation, Marti, Puertas [4] discovered that the more complicated the transit of commodities, the more important the logistics indicators with trade facilitation is especially important for Middle Eastern exporters. Marti, Puertas [4] further revealed that while the index is more essential for exporting countries, it is less important for importing countries' trade flows. Further, the logistical performance of the countries within this region has more impact on the export volumes in comparison the logistical performance of the importing countries [33]. In terms of the sub components of LPI, timeliness component is the most important for this region.

## Africa

The foreign policy is not a fundamental issue for many poor countries in Africa, but its geographical location highly is. This is because many African countries are geographically handicapped. It is noteworthy that these countries have unique geographical features while some countries are located far from the sea further distancing them from accessing seaports for international trade. Due to its variations in locations in African countries, these nations had to face high transportation costs to enter the world market [34]. Oil rents have a negative impact on growth in Algeria, Angola, Egypt, and Libya, whereas net oil exports have a negative impact on GDP growth in Africa, Angola, Egypt, Libya, and Algeria in the medium and long term, but are favourable in Nigeria [35]. Economic activity in South Africa over the past 40 years has been sluggish due to lack of investments and policies to open the country to international trade. In the early 1980s, per capita GDP peaked at historic highs and then fell to moderate growth after the political transition of the early 1990s. But with these recent developments, by 2004, per capita GDP had increased around 40% against that achieved in 1960 [36].

The LPI score for African countries is the lowest, especially in aspects of trade and transportation infrastructure, as well as customs and border clearance [37]. Improving any LPI component in African countries can lead to significant growth in exports and respectively the export share of African goods in global trade can be improved [37]. This analysis has been conducted by only considering the global trade performance statistics of countries in 2016. From the 6 sub-components of LPI, only two "competence of logistics services" and "quality of trade and

transport-related infrastructure" have a weak, yet positive correlation with GDP per capita: indicator of economic growth [38]. In reality, the ability of Sub-Saharan Africa nations to link to global value chains is heavily influenced by the regional dimension of infrastructure and trade facilitation policies [39]. Thus, improving logistics performance in terms of infrastructure can have a positive impact on exports and trade facilitation across Africa.

## The Americas

The tariffs and non-tariff measures in Brazil, Colombia, Uruguay, and Venezuela, time to import (almost double that of competitors on average), cost to import (on average 26% higher than other destinations), and poor English proficiency in business circles in most South American countries are the most common barriers that South African exporters face in South America [40]. The study carried out by Bazani, Pereira [41] revealed that in comparison to other countries, Brazil has a middling logistics performance. Its worst performance was in the area of customs, while its finest was in the area of punctuality. Furthermore, industrialised countries are disproportionately represented in the cluster with the best logistical performance. Martí, Puertas [33] has found that the within the South American continent the logistic performance of their own countries bears greater weight on the export volumes, as opposed to the logistics performance of the importing country. Further, LPI sub-components of international shipments as well as customs and tracking are highly significant with trade within the South American region.

## Global

The influence of logistics performance on exports and imports across countries worldwide has been investigated in few studies using an income-level approach. [42] reveals that there are gaps in overall LPIs between high-income economies and low and middle-income economies, and this has aggravated slightly since 2007 in low- and high-income economies [42]. Further, Celebi [29] found out that depending on a country's income level the impact of logistics performance on exports and imports vary. While LPI Physical infrastructure has the highest impact on exports in low-middle-income economies, logistics performance generally has a major impact on imports in upper-middle-income economies such as China, Mexico, Thailand, and Turkey [29]. In order to improve the logistics performance index in Africa, Asia, and the EU, infrastructure, human factors, and institutions are required [43]. In Europe, the human factor is significantly more important for improving the LPI over time, whereas in Asia, infrastructure remains key. The development of Africa's logistics is dependent on these all three elements [43].

Additionally, recent studies justify that overall logistics performance is positively and statistically significantly linked to exports and imports with a positive correlation [42,44]. Moreover, using econometric research, Chakraborty and Mukherjee [45] sought to determine the relationship between logistics performance and exports in higher- and lower-income nations in 2007, 2010, 2012, and 2014. The findings revealed that lower-income countries require aid for trade facilitation. Korinek and Sourdin [46] found out that LPI has a considerable impact on commerce in most middle-income countries where infrastructure development is a priority. Among countries with similar levels of income, economies with better logistics performance recorded additional GDP growth of 1% and trade growth of 2% [9]. Therefore, it can be concluded that the LPI is a good indicator of a country's participation in global value chains [47].

While the LPI considers a variety of factors that influence a country's logistics performance, such as infrastructure, information technology, service quality, government regulations, and policies etc., to measure the overall performance of the industry or nation in green logistics

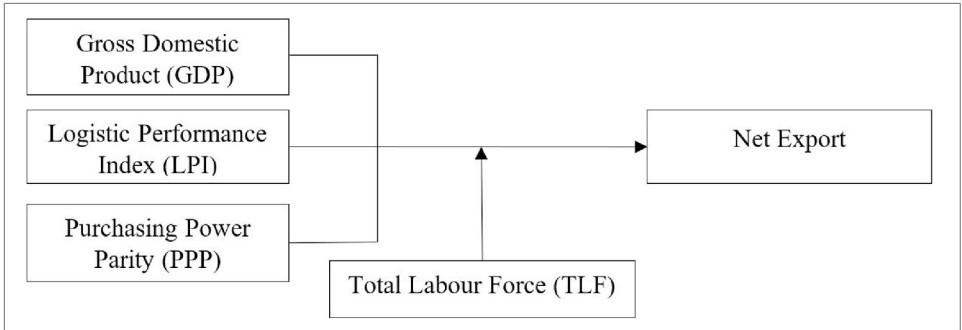

**Fig 1. Conceptual framework interrelationship of variables.** Source: Authors' demonstration based on literature.

(GL) activities, the green logistics performance index (GLPI) considers resource investment, adoption of cutting-edge technology, and adherence to environmental legislation, among other factors. Similar to LPIs, GLPI and its core index variables are structured using several key categories of GL activities and is a key area of exploration in previous studies [48].

Although various research on the impact of LPI on international trade has been undertaken on a worldwide scale, no comprehensive study including all continents globally has been discovered in the literature. Examined research so far has certain limitations in terms of geographical coverage, time duration, and comparative analysis conducted among regions or continents. This research article is expected to fill the above-mentioned research gap by providing a comparative examination of the importance of the GDP and the LPI in international trade across all continents and Middle East countries.

A conceptual framework was built upon the foundation of the literature discussed and is presented in Fig 1.

## Methodology

The study employed panel data across the years 2007, 2010, 2012, 2014, 2016, and 2018 of 142 countries. The LPIs has been published by the World Bank biennial since 2007, which resulted in the gaps in the years analyzed. Specifically, 37 Asian, 41 European, 41 African, 3 Oceanian, 11 North American, and 9 South American and 14 Middle East countries were studied. The countries classified as the Middle East are Bahrain, Cyprus, Egypt, Iraq, Iran, Israel, Jordan, Kuwait, Lebanon, Oman, Qatar, Saudi Arabia, Turkey, and Yemen. Table 1 presents the summary of descriptive statistics and Table 2 shows the data sources and definition of variables.

The purpose of this study is to examine the impact of GDP and LPI on international trade, which determines the country's NEX. Two static linear panel models are developed based on the conceptual framework and the literature review. These models were also well endorsed in past literature [49]. Using LPI, GDP, PPP, and Total Labour Force (TLF) as inputs, an empirical model was developed for the NEX as follows:

$$NEX_{it} = \beta_0 + \beta_1 \text{GDP}_{it} + \beta_2 \text{LPI}_{it} + \beta_3 \text{PPP}_{it} + \varepsilon_{it} \qquad (1)$$

$$NEX_{it} = \beta_0 + \beta_1 GDP_{it} + \beta_2 LPI_{it} + \beta_3 PPP_{it} + \beta_4 (GDP \times TLF)_{it} + \beta_5 (LPI \times TLF)_{it} + \beta_6 (PPP \times TLF)_{it} + \varepsilon_{it} \qquad (2)$$

Eq (1) is created to test the impact of GDP, LPI, and PPP on the net export in Asian, European, African, Oceania, North America, South America, Middle East, and Global countries. Eq (2) further extends this by incorporating terms (GDP x TLF), (LPI x TLF), (PPP x TLF) to

**Table 1. Summary descriptive statistics for the key variables.**

| Countries | | NEX (billion) | GDP (billion) | LPI | PPP | (GDP×TLF) (quintillion) | (LPI×TLF) (billion) | (PPP×TLF) (billion) |
|---|---|---|---|---|---|---|---|---|
| | | | | | | **Variables** | | |
| Global | Obs. | 796 | 796 | 796 | 796 | 796 | 796 | 796 |
| | Mean | 3.3196 | 765.0088 | 2.9276 | 280.7232 | 157.566 | 0.0731 | 8.9027 |
| | Std. Dev. | 64.1859 | 2,276.513 | 0.5745 | 1,080.408 | 1170.554 | 0.2765 | 55.6921 |
| | Min. | -720 | 1.5599 | 1.21 | 0.1 | 0.0002496 | 0.0003 | 0.000096 |
| | Max. | 309.98 | 20,128.6 | 4.23 | 16,945.5 | 15,594 | 2.8804 | 630.961 |
| Asia | Obs. | 207 | 207 | 207 | 207 | 207 | 207 | 207 |
| | Mean | 15.2604 | 1239.397 | 2.9147 | 634.0725 | 481.8676 | 0.1782 | 31.0210 |
| | Std. Dev. | 53.4295 | 2973.859 | 0.5582 | 1,951.993 | 2213.159 | 0.5132 | 106.1363 |
| | Min. | -122.91 | 4.5990 | 1.21 | 0.1 | 0.0010 | 0.0003 | 0.000108 |
| | Max. | 309.98 | 19,840 | 4.19 | 16,945.5 | 15,594 | 2.8804 | 630.961 |
| Europe | Obs. | 238 | 238 | 238 | 238 | 238 | 238 | 238 |
| | Mean | 15.9529 | 628.1848 | 3.3193 | 10.6340 | 19.7486 | 0.03120 | 0.0687 |
| | Std. Dev. | 48.5027 | 1,029.215 | 0.5399 | 28.6189 | 53.6053 | 0.0476 | 0.2521 |
| | Min. | -85.0599 | 10.3174 | 2.08 | 0.1 | 0.0018 | 0.0005 | 0.00009 |
| | Max. | 260 | 4,288.53 | 4.23 | 140 | 295.175 | 0.2037 | 1.8309 |
| Africa | Obs. | 215 | 215 | 215 | 215 | 215 | 215 | 215 |
| | Mean | -1.7634 | 124.9463 | 2.4961 | 297.2722 | 3.0934 | 0.0227 | 1.8791 |
| | Std. Dev. | 8.2592 | 238.7026 | 0.3320 | 539.2544 | 9.2419 | 0.0300 | 3.6939 |
| | Min. | -31.77 | 1.5599 | 1.61 | 0.4 | 0.00024 | 0.00039 | 0.0005 |
| | Max. | 84.55 | 1219.86 | 3.78 | 3,159.3 | 59.6015 | 0.1531 | 20.9289 |
| Oceania | Obs. | 17 | 17 | 17 | 17 | 17 | 17 | 17 |
| | Mean | -3.0023 | 471.7638 | 3.3129 | 1.2941 | 5.1691 | 0.0196 | 0.0076 |
| | Std. Dev. | 8.0690 | 534.0586 | 0.6393 | 0.2656 | 7.0260 | 0.0204 | 0.0079 |
| | Min. | -27.5 | 9.0019 | 2.24 | 0.9 | 0.0034 | 0.00083 | 0.0003 |
| | Max. | 5.46 | 1319.47 | 3.88 | 1.5 | 17.483 | 0.0496 | 0.0198 |
| North America | Obs. | 65 | 65 | 65 | 65 | 65 | 65 | 65 |
| | Mean | -58.5661 | 2072.544 | 2.9518 | 40.5449 | 280.4887 | 0.0844 | 0.1467 |
| | Std. Dev. | 166.8328 | 5210.124 | 0.4973 | 97.1284 | 843.9249 | 0.1794 | 0.2331 |
| | Min. | -720 | 11.4186 | 2.25 | 0.5 | 0.0022 | 0.00055 | 0.00018 |
| | Max. | 31.96 | 20128.6 | 3.99 | 363.4 | 3330.88 | 0.6485 | 0.8614 |
| South America | Obs. | 54 | 54 | 54 | 54 | 54 | 54 | 54 |
| | Mean | -1.4135 | 616.3905 | 2.8183 | 427.7926 | 36.8712 | 0.0599 | 4.3554 |
| | Std. Dev. | 12.8868 | 871.4718 | 0.2526 | 759.0121 | 90.7008 | 0.0878 | 9.0905 |
| | Min. | -65.37 | 47.76229 | 2.25 | 0.4 | 0.0792 | 0.0041 | 0.0026 |
| | Max. | 23.83 | 3140.61 | 3.32 | 2528.1 | 317.451 | 0.3167559 | 34.8178 |
| Middle East | Obs. | 78 | 78 | 78 | 78 | 78 | 78 | 78 |
| | Mean | 13.3896 | 497.4983 | 2.9575 | 629.7551 | 9.3685 | 0.0254 | 14.4831 |
| | Std. Dev. | 41.4903 | 570.8066 | 0.3614 | 2,446.388 | 15.8857 | 0.03148 | 66.3335 |
| | Min. | -36.18 | 23.1820 | 2.11 | 0.1 | 0.0138 | 0.0016 | 0.00010 |
| | Max. | 187.63 | 2222.15 | 3.66 | 16,945.5 | 72.953 | 0.1054 | 470.577 |

determine the moderating effect of TLF on the relationship between GDP, LPI, PPP on the net export in Asia, European, African, Oceania, North America, South America, Middle East, and Global countries.

$NEX_{it}$ represents Net Exports, where Country is denoted by $i$ and the time is denoted by $t$. $GDP_{it}$, $LPI_{it}$ and $PPP_{it}$ represents Gross Domestic & Product, LPI, Purchasing Power Parity

**Table 2. Data sources and definition of variables.**

| Variable | Definition | Source |
|---|---|---|
| NET | Net Export (US$) | World Bank Reports |
| GDP | Gross Domestic Product (US$) | Federal Reserve Economic Data |
| LPI | Logistics Performance Index (Scaled 1 to 5) | World Bank Reports |
| PPP | Purchasing Power Parity (LCU per International $) | Knoema.com |
| (GDP × TLF) | GDP× Total Labour Force | |
| (LPI × TLF) | LPI × Total Labour Force | |
| (PPP × TLF) | PPP × Total Labour Force | |

respectively. (GDP × TLF) $_{it}$ denotes the moderate effect of GDP and TLF, (LPI × TLF)$_{it}$ denotes the moderate effect of LPI and TLF, (PPP × TLF)$_{it}$ denotes the moderate effect of PPP and TLF. The error term is represented by $\varepsilon_{it}$.

The variables were applied to a standardisation procedure. The coefficients associated with each variable were modified properly for the disparity in variable sizes throughout the standardization procedure. The data was then transformed into a robust standard error to reduce the problem of heteroscedasticity.

## Estimation procedures

Due to the short duration of time series, dynamic and non-stationary panel data techniques are problematic. Static panel data techniques, such as the pooled ordinary least square (POLS) model, random-effect model, and fixed effect (FE) model, are used to evaluate the given regression model for international trade, according to econometrics literature. The estimated model's characteristics are: country-specific impact and time-specific impact, identify the best acceptable model. Forecasts utilizing the RE and FE models are favored over the POLS model if a country-specific impact occurs, wherein the drift does not vary over time among countries. In estimating, the RE model outperforms the FE model if variations in the intercepts are arbitrary. When neither influence occurs, the coefficients in Calculations (1) and (2) must be estimated using the POLS model (2). To select one of the three approaches for evaluating the effects of GDP and LPI on international trade, a series of specification tests must be conducted. To evaluate if the POLS or FE model must be utilised, the poolability F-test is being initially used, accompanied by the Breusch-Pagan Lagrange Multiplier (LM) test. If the poolability F-test and the Breusch-Pagan LM test both show that only POLS are better than in other models, POLS estimation should be used. The Hausman test, on the other hand, will be utilised to choose among RE and the FE models.

## Empirical results and discussion

According to the Average LPI graph under Fig 2, the highest LPI score is shown in the European and Oceanian continents under the analysis. These continents consist of countries that have strong economic conditions of their own. As an example, countries such as New Zealand, Australia are placed among the top 25 countries in line with the LPI rankings in the world. European countries are well placed and most of the EU member states are on top of the LPI index. Therefore, the European continent follows Oceanian closely in terms of logistics performance development over the last decade. Similarly, Sergi, D'Aleo [43] also found the closeness of the LPI values of Oceania and EU through their descriptive analysis. As Middle East countries have more resources that could be exported, the GDP of Middle East countries are higher compared to other continents. As a result, shown in the below Table 3 according to the RE

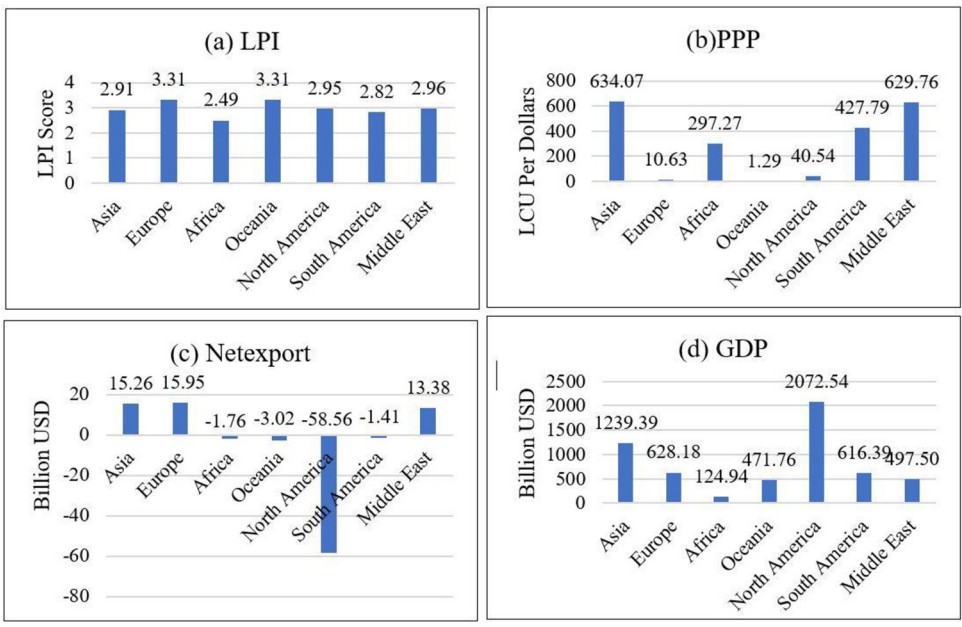

**Fig 2. Average of the variable by geographical area from 2007 to 2018.** Source: Authors' Illustration based on the data.

model, there is a strong relationship and the significance between NEX and the GDP indicates a 1% significance level; this means 99% of accuracy could be achieved between the two variables. Here, African continent records the lowest LPI score among all continents as most of its countries have fragile economies severely affected by several hitches.

As illustrated in the average PPP graph, Asian continent countries must pay more attention to buy the same goods or services in the domestic market accordingly with the US. A similar condition can be observed in Middle East countries as well. There is a sizeable difference between North and South America concerning PPP. South America has a higher local currency unit (LCU) value even compared to the African continent. As Oceania has strong economic conditions in its countries, it has the lowest LCU in the PPP aspect.

The average GDP graph illustrates that how the GDP varies across continents around the globe. The highest GDP value is indicated by the North American continent as it has several

**Table 3. Specification tests for the panel model selection.**

| Tests | $H_0$: pooled OLS | $H_0$: pooled OLS | $H_0$: random effects |
|---|---|---|---|
|  | $H_1$: fixed effects | $H_1$: random effects | $H_1$: fixed effects |
| Eq (1) |  |  |  |
| Poolability *F*-test | 7.19*** |  |  |
| Breusch-Pagan LM test |  | 1735.49*** |  |
| Hausman test |  |  | 3.47*** |
| Eq (2) |  |  |  |
| Poolability *F*-test | 6.87*** |  |  |
| Breusch-Pagan LM test |  | 1428.63*** |  |
| Hausman test |  |  | 22.49*** |

Note
*** Significant at 1% level.

strong economic nations. Though the Asian continent has fragile economic conditions in many countries, it also has several countries that hold steady and strong economies. As such, still, the GDP is somewhat considerable in Asia compared to many continents. Europe, South America, and Middle East continents having middle level GDPs and Oceania has considerably lower GDP levels though the countries taken into consideration have strong economies. As with every other variable, the African continent indicates significantly lower economic conditions within the continent even under GDP levels.

As the Average Net Export graph shows, Asia has the highest exports in the years under consideration for the analysis. In Asia, countries such China, Japan produce a wide range of products in higher volumes to export. For this reason, facilitating the exchange of good via import and export trade, the East–West maritime route is much popular in the logistics industry. Europe and the Middle East also consist of high-income export economies (known as rich economies) and international trade performance is relatively stronger in this region. Though the African continent has poorer conditions in their economies, they still have some exports such as oil and minerals, including copper and iron as well as agricultural products, including the cotton and cocoa [43], compared to those of other continents such as Oceania, South and North Americas. Among these, North America has a greater minus value on their exports thus signals that these economies consume more than what they produce. Yet, economic conditions in North America is stronger than a few other continents globally.

The results of the specification tests for Eqs (1) and (2) are shown in Table 3. The poolability F-test findings and the Breusch-Pagan LM test indicate that the null hypothesis of the POLS model as the preferred specification is rejected at a 1% significance level. Based on these specification tests, the POLS model is determined as less appropriate for the current investigation. Thus, prior to estimating the coefficients, the Hausman test is required to be carried out to select between RE and FE models. The test rejects the null hypothesis of the FE model, implying that the intimation outcomes of the RE model are more efficient than those of the FE model. Based upon this notion, the RE model estimation was used for a more in-depth analysis of Eq (2) and reported in Table 4.

The results in Table 4 above depicts the standardized coefficients and standard errors for global countries and the continents separately, to determine the influence of GDP, LPI, and PPP on NEX. The TLF acts as the moderating variable to improve the significance level of each independent variable (GDP, LPI, and PPP). In the global context, the global NEX is affected by both GDP and LPI at a 1% of significance level while GDP has a negative impact and LPI has a positive impact on NEX. By moderating the significance level of independent variables, LPI could be brought towards 5% of significance level from 10%, with a higher coefficient value. Therefore, it is reasonable to conclude that LPI has a significant relationship with a higher coefficient value towards NEX in the global context compared to the other two independent variables, i.e. GDP and PPP. Since the estimated coefficients of LPI is positive and highly significant, we can conclude that an increase in LPI will result in a surge in the net exports of a country irrespective of the geographical region concerned. Marti, Puertas [4] also support this finding that the LPI is significant and positively related to exports in emerging economies.

When focusing continent wise, similar to the global context, the Asian continent shows a significant relationship between GDP, LPI and NEX. However, the LPI has a higher significance level compared to GDP towards NEX. The Asian continent has a mix of strong and steady as well as fragile economies, both of which results in a more positive effect towards NEX in the continent.

Only LPI is significant in the European continent. All in all, European countries are the top performers in LPI, having their dominant supply chain industries [11]. Moreover, many European countries are within the top LPI quintile which includes top-performing countries,

**Table 4. The results of the random effect model.**

|  | Global countries | Asian countries | Europe countries | Africa countries | Oceania countries | North America countries | South America countries | Middle East countries |
|---|---|---|---|---|---|---|---|---|
| Variable | NEX (RE) | NEX (RE) | NEX (RE) | NEX (RE) | NEX (RE) | NEX (RE) | NEX (RE) | NEX (RE) |
| GDP | -0.0252*** | -0.0183* | -0.02633 | -0.0250* | 0.0851* | -0.0079 | -0.0149 | 0.1333*** |
|  | (0.0085) | (0.0106) | (0.0463) | (0.0131) | (0.0472) | (0.0255) | (0.0136) | (0.0099) |
| LPI | 8.9509*** | 2.5710*** | 1.5510** | -1.5909 | 1.6810** | 2.4910 | 5.7409 | -4.7609 |
|  | (3.0809) | (7.9109) | (6.6109) | (1.7809) | (7.3609) | (2.8310) | (6.5509) | (8.1209) |
| PPP | -1496885 | 36928.78 | 9.9507 | -1125405 | -3.4210** | -5.2408** | 4179954** | -1.9907 |
|  | (1008834) | (1225265) | (1.3808) | (1348543) | (1.6210) | (2.4408) | (1723415) | (1.2607) |
| (GDP × TLF) | 1.2411 | 1.8911 | 1.0809** | 6.6010*** | -2.0209 | -2.0611 | -3.1610 | -2.6809** |
|  | (1.1911) | (1.6211) | (4.9510) | (2.0810) | (1.4309) | (1.6310) | (2.0010) | (1.2209) |
| (LPI × TLF) | 137.2198** | 62.8451 | 166.5724 | -10.6089 | -8611.443*** | -695.4385*** | 405.7024 | -1133.745* |
|  | (57.3295) | (94.5058) | (948.877) | (98.4242) | (1383.399) | (238.5306) | (333.1358) | (589.92) |
| (PPP × TLF) | 0.0145 | -0.0213 | -51.56696 | 0.0919 | 17450.18*** | 233.4718** | -0.9340*** | 0.7805* |
|  | (0.0404) | (0.0501) | (33.0750) | (0.2886) | (1906.223) | (107.1303) | (0.3522) | (0.4587) |
| Constant | -1.5610** | -5.6610*** | -4.3210* | 3.5809 | -8.5409*** | -6.4110 | -1.8710 | 1.6310 |
|  | (7.3609) | (1.8710) | (2.2210) | (3.9709) | (3.3009) | (7.3010) | (1.5210) | (2.6810) |
| Obs (N×T) | 796 | 207 | 238 | 215 | 17 | 65 | 54 | 78 |
| No. of country (N) | 142 | 37 | 41 | 41 | 3 | 11 | 9 | 14 |
| No. of year (T) | 6 | 6 | 6 | 6 | 6 | 6 | 6 | 6 |
| R-Squared-within | 0.027 | 0.011 | 0.074 | 0.028 | 0.782 | 0.053 | 0.299 | 0.002 |
| between | 0.460 | 0.382 | 0.436 | 0.429 | 1.000 | 0.998 | 0.818 | 0.874 |
| overall | 0.433 | 0.312 | 0.399 | 0.197 | 0.869 | 0.970 | 0.371 | 0.613 |

Note: (.) indicates the Robust standard error

***, **, * Significant at the 0.01, 0.05 and 0.10 level respectively.

whereas the majority are in the high-income group. However, by moderating the GDP coefficient with the moderate variable, a higher level of significance is achieved. Larger the economy, anticipated level of imports and exports are also higher resulting in a positive relationship between GDP and international trade [4].

The African continent does not show any significant relationship in either LPI or PPP with NEX. But shows a significant negative relationship with GDP. In the past four decades, African economies have stagnated and become one of the poorest regions, while other developing countries have seized growth opportunities [50]. African countries face severe logistics constraints and lie at the bottom of the aggregate LPI rankings globally; this region also encounter hardships in connecting with global supply chains, given that most countries within the continent have fragile economies affected by conflict, natural disasters, political unrest or being landlocked [11,43]. Thus, the poor performance of the African economies is evident from the results of this study.

The Oceanian continent is unique with all coefficients of independent variables being significant towards NEX. Although significant, the impact from PPP is understandably negative towards NEX. Since of poverty, taxes, and other transaction costs, PPP can hinder trade between two nations because PPP inflation and exchange rates may diverge from market exchange rates. Australia, Fiji and New Zealand are key contributors to the world economy and when considering LPI rankings, Australia and New Zealand are placed among the top 25 in the world and fall within the second LPI quintile [11]. Similar to the European continent,

the high performance in logistics could have been in favour of LPI to be a significant contributor to NEX.

Only PPP is significant within the North and South American continent. Although the North American continent shows a positive impact of PPP towards NEX similar to Oceania, it is the opposite scenario for the South American countries. By moderating the coefficients through moderate variables does not show any significant level in GDP and LPI in South America while North America shows a positive 1% of significance. The study conducted by Mendes dos Reis, Sanches Amorim [51] on the 'Impact of Logistics Performance on Argentina, Brazil, and the US Soybean Exports from 2012 to 2018: A Gravity Model Approach' reveals the importance of considering the LPI throughout several indicators instead of aggregating at the country level. These scholars concluded that some indicators could affect positively while the other indicators can affect negatively or still be not significant to trade, at least in the soybean case. Therefore, this argument is supported by the authors of the present study for not having any significant value for coefficients like LPI. Main countries in the South American region have fallen to the 1st quarter of the LPI defined by the World Bank in 2018.

In the Middle East context, the only significant coefficient is GDP with a positive impact and 1% of the significance level. By moderating these three variables (GDP, LPI and PPP), these could become significant at a 5–10% level of significance. Results of this study show an impact on GDP is relatively less yet positive confirmed with the findings of Erkisi [32] who asserted that a long–term relationship exists between NEX and GDP.

Therefore, it can be concluded that GDP coefficients are significant in the global context concerning Asian, Oceanian, African and Middle East continents. LPI has a significant relationship in the global context, concerning Asian, European, and Oceanian continents. Further, PPP is being significant in the Oceanian, North America, and South America continents where latter has a higher coefficient value compared to coefficient values of other regions.

## Conclusion & policy implications

Despite the vast number of developed empirical papers on international trade, only a few publications have used panel data analysis. According to the information available to authors, no studies have been conducted examining countries across all continents. Thus, one of the main empirical contributions of this paper is to present the combined impact of LPI, GDP and PPP globally.

The paper examined the impact of GDP and LPIs on international trade on other continents relative to global countries. Static panel data techniques such as the pooled ordinary least square (POLS) model, RE model, and FE model were used to evaluate the given regression model for international trade.

This study has drawn several main insights from the results obtained. Our results reveal the importance of considering the NEX throughout their several indicators GDP, LPI and PPP instead of aggregating at the country level. We conclude the possibility of mixed outcomes, where some indicators can affect positively while others can affect negatively or still be not significant in international trade. Countries in the African continent have highly fragile economic conditions which imply that LPI have no significance towards the continent's NEX. Meanwhile, in some economies, GDP and LPI show higher significance levels towards international trade specially for NEX. However, GDP has a significant impact. For example, in the Oceanian continent, countries like New Zealand, Australia and Fiji contribute to world international trade on a larger scale. Countries' efforts to improve logistics performance indirectly affect the growth of international trade. This issue is crucial from the international aspect and development of these countries. Additionally, the results of previous research regarding the effects of the size of the economy and the distance between trading partners have been confirmed [1].

In other words, it is evident from research-based studies that stronger the economies, higher and better the LPI and GDP values. Additionally, the scenario of the African continent indicates that economic instability or fragile economies are likely to result in poor LPI indicators. Arguably, the status of the economy seems to heavily influence LPI and GDP, hence, with a likelihood for these indicators to be overrated and underrated. Hence, in such circumstances, researchers need to be vigilant to determine a different mix of indicators as realistic measures to be used to measure the performance of international trade.

Considering the scenario of soybean where LPI is influenced, on the one hand, this means that the coefficients associated with LPIs can vary based on the export basket of a given country. On the other hand, LPI may be influenced by the import mix as well as the export-import mix. As mentioned under Section 2 (Literature Review), industrialised countries tend to be disproportionately represented in the cluster with the best logistical performance. All in all, it means that the LPI can be influenced by the type of country (fragile or strong economy) which the importer or exporter, the bargaining power of the importer and exporter and the demand for the goods traded and the PPP.

The study findings on the impact of the GDP and LPI on international trade are vital for the government in these geographical regions to implement new legislation and make decisions on trade-related policies. This paper contributes to logistics policy implications for continents dealing with several geographical locations of each continent and can guide to strengthen their economies regardless of whether steady or fragile. While both Global and Asia have a significant relationship with both GDP and LPI, EU continent only shows a significance towards LPI. Moreover, while Oceanian has significance in all three variables, Africa has significant impact on GDP. PPP is significant in both American continents with a significance in GDP for the Middle East continent the decision and policymaking process could accelerate by improving those variables impacts to the international trade.

According to empirical findings, logistics performance seems to play a significant role in fostering export performance, strengthening policy implications and enhance the logistics performance of countries.

Even though a large sample of data, when analyzing each continent through their LPI, GDP and PPP parameters, some countries were excluded (Botswana, Ethiopia, Morocco, Mozambique, Uganda, Yemen and etc.) due to the lack of published data.

For further studies, the empirical analysis could be extended country wise, considering the rest of the countries of each continent with changing LPI values over time. The list of variables was established based on previous studies. However, future research could determine which variables are more relevant and influential (in what continents and regions) and hence adequate to measure with logistics performance, GDP and PPP using empirical data. To gain broader insights on various perspectives concerning international trade, future studies can focus on the variations between the LPI sub-indices such as customs, infrastructure, logistics, timelines, international shipments, and tracking and tracing.

## Supporting information

**S1 Data.**
(XLSX)

## Acknowledgments

The authors would like to thank Ms. Gayendri Karunarathne for proof-reading and editing this manuscript.

## Author Contributions

**Conceptualization:** Ruwan Jayathilaka, Chanuka Jayawardhana, Nilupul Embogama, Thisara Gamage, Nethmali Kuruppu.

**Data curation:** Chanuka Jayawardhana, Nilupul Embogama, Thisara Gamage, Nethmali Kuruppu.

**Formal analysis:** Ruwan Jayathilaka, Chanuka Jayawardhana, Nilupul Embogama.

**Investigation:** Chanuka Jayawardhana, Nilupul Embogama.

**Methodology:** Ruwan Jayathilaka, Chanuka Jayawardhana, Nilupul Embogama.

**Resources:** Ruwan Jayathilaka.

**Software:** Chanuka Jayawardhana, Nilupul Embogama.

**Supervision:** Ruwan Jayathilaka.

**Validation:** Chanuka Jayawardhana, Nilupul Embogama, Shalini Jayasooriya, Navodika Karunarathna.

**Visualization:** Chanuka Jayawardhana, Nilupul Embogama, Shalini Jayasooriya, Navodika Karunarathna, Thisara Gamage, Nethmali Kuruppu.

**Writing – original draft:** Ruwan Jayathilaka, Chanuka Jayawardhana, Nilupul Embogama, Shalini Jayasooriya, Navodika Karunarathna.

**Writing – review & editing:** Ruwan Jayathilaka.

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
