## [Decision Letter · Decision Letter 0]

21 Dec 2021

PONE-D-21-35917Gross Domestic Product and Logistics Performance Index Drive the World Trade: A Study based on All ContinentsPLOS ONE

Dear Dr. Ruwan Jayathilaka,

Thank you for submitting your manuscript to PLOS ONE. After careful consideration, we feel that it has merit but does not fully meet PLOS ONE’s publication criteria as it currently stands. Therefore, we invite you to submit a revised version of the manuscript that addresses the points raised during the review process.

This article is interesting but some information is hard to follow as mentioned by reviewer 1. Please address your idea clearer in the revision.

We look forward to receiving your revised manuscript.

Kind regards,

Wen-Wei Sung, M.D., Ph.D.

Academic Editor

PLOS ONE

Journal Requirements:

Reviewers' comments:

Reviewer's Responses to Questions

**Comments to the Author**

1. Is the manuscript technically sound, and do the data support the conclusions?

Reviewer #1: Partly

Reviewer #2: Partly

2. Has the statistical analysis been performed appropriately and rigorously? 

Reviewer #1: N/A

Reviewer #2: Yes

3. Have the authors made all data underlying the findings in their manuscript fully available?

Reviewer #1: No

Reviewer #2: Yes

4. Is the manuscript presented in an intelligible fashion and written in standard English?

Reviewer #1: No

Reviewer #2: Yes

5. Review Comments to the Author

Reviewer #1: The author has studied the relation between GDP and LPI across continents, they use random effect regression model to check the relation between export, GDP and LPI in Asia, Europe, Oceania, Middle East, America and Africa etc. They find that LPI has a positive relation with net exports globally and specifically within the continents of Asia, Europe, and Oceania. They also find many other interesting conclusion.

but there are still some problems, please check very carefully:

1. There are some sentences that I can't get the point, like "Logistics acts as the backbone of trade, with the GDP and PPP making significant contributions to the economic growth of economies. " GDP is an index that measure the gross domestic product of a country/region, PPP is a way of measure foreign exchange.

Please check the whole paper if there are other sentence like this.

2. Please reorganize your section, for example in literature review, Africa, Asia, and the EU appeared after Africa, Asia, EU section. South African appeared after Africa.

3. In Fig2,

don't let xticklabel cover the number that others can't recoginize the number.

4. Consider add more control variable in your regression, and it should be more convincible.

5. When study the Middle East, the oil production country, or other economics with large size of economic growth based on one economic sector, the author should consider whether the oil price has strong impact on export or GDP.

Reviewer #2: The paper investigates a really interesting and timely topic. The coherence between GDP and LPI is a unique relationship and can answer several research questions. The abstract is appropriate and compact as it is. The tile is acceptable as well; the authors try to follow it and more-or-less succeeded. However, when it comes to literature review, the global approach can have disadvantages: the subparts are very short and not comprehensive enough. We see a quick run over the continents, however, a very draft and inappropriate analysis only. So, I recommend the extending of the literature wher international sources are processed in a critical, analytical and comprehensive way.

6. PLOS authors have the option to publish the peer review history of their article (what does this mean?). If published, this will include your full peer review and any attached files.

Reviewer #1: No

Reviewer #2: No

---

## [Author Response · Author response to Decision Letter 0]

3 Feb 2022

Point–by–point response to reviewers

Comments from Authors: Please note that page numbers and line numbers refereed in this document is align with the revised manuscript which has track changes.

Comments of Reviewers:

Reviewer #1: The author has studied the relation between GDP and LPI across continents, they use random effect regression model to check the relation between export, GDP and LPI in Asia, Europe, Oceania, Middle East, America and Africa etc. They find that LPI has a positive relation with net exports globally and specifically within the continents of Asia, Europe, and Oceania. They also find many other interesting conclusion.

but there are still some problems, please check very carefully:

Comments of Authors:

Thank you very much for the comment.

Comments of Reviewers:

There are some sentences that I can't get the point, like "Logistics acts as the backbone of trade, with the GDP and PPP making significant contributions to the economic growth of economies. " GDP is an index that measure the gross domestic product of a country/region, PPP is a way of measure foreign exchange.

Please check the whole paper if there are other sentence like this

Comments of Authors:

Comment has been noted and this has been corrected in the revised manuscript. The sentence has been deleted and the paragraph has been adjusted.

“The literature pertaining to international trade flows, repeatedly stresses that the logistics performance, GDP and the PPP have an impact on the volume of international trade. Logistics has been featured as a critical factor in the facilitation of trade and in turn, a stimulator of a nation’s economic development. Thus, it significantly influences bilateral trade flows [1, 6, 7]. The LPI is a comprehensive index designed to assist countries in identifying challenges and opportunities in trade logistics work evidence [8-11]. Further, previous studies highlight that a rise in the GDP of the trading partners escalates the export trade volumes [12, 13]. However, the literature indicates a unidirectional causality from export to import, meaning that in the long run, export leads to import but not vice versa [14]. Furthermore, exchange rates are at the core of scholarly work related to international trade, and exchange rates are said to adjust at a level set as per PPP [15]. A discussion of existing literature pertinent to the variables studied in this research are discussed continent wise henceforth.”

(Page 4, Lines 14 -30).

Comments of Reviewers:

Please reorganize your section, for example in literature review, Africa, Asia, and the EU appeared after Africa, Asia, EU section. South African appeared after Africa.

Comments of Authors:

Comment is well received. 

The section on Africa, Asian & EU is shifted to the global section. (Page 8, line 29- Page 9, line 2)

The new subs sections under the literature review are Asia-Pacific, EU, Middle East, Africa, The Americas and Global. We have revised the literature review and added 09 new research articles as described in detail in Point–by–point response to Reviewer 2. (Comment 2, Page 5),

Comments of Reviewers:

In Fig2, don't let xticklabel cover the number that others can't recognize the number.

Comments of Authors:

Thank you. This has been corrected as follows in figure: (Page 14)

Comments of Reviewers:

Consider add more control variable in your regression, and it should be more convincible.

Comments of Authors:

We appreciate your comment. In the initial analysis of our paper, we considered the following control variables for each country such as Total Population, Population Ages 65 and above, Population ages 0-14, Land area, Presence of a coastline, Presence of Covid19.

However, these variables weren’t significant in the initial analysis and we have observed that these create a multicollinearity issue. Further, in the presence of these control variables, the independent variables become insignificant. Therefore, when selecting the final model, we did not include these as control variables.

The authors can provide the generated results, if required.

Comments of Reviewers:

When study the Middle East, the oil production country, or other economics with large size of economic growth based on one economic sector, the author should consider whether the oil price has strong impact on export or GDP.

Comments of Authors:

Thank you for the comment. It was carefully considered, however, there is no single source which has publicly accessible data on changes in oil prices for all the years 2007, 2010, 2012, 2014, 2016, and 2018 for each country which was considered in this study. If the oil price was to be considered, the data will have to be gathered from multiple sources, which poses questions to validity and reliability. 

Further, the sample considered will drop significantly by approximately 78%. We have considered data of 142 countries across 6 years in our study, but data on changes in oil prices across the years we have used is available only for 28 countries, therefore we will have to approximately remove 684 observations and our dataset will be reduced 168 observations. (28 countries * 6 years = 168). 

However, we highly appreciate the comment, and the authors believe that this will be very good variable to incorporate to a future study which focuses on middle east countries.

Comments of Reviewers:

The paper investigates a really interesting and timely topic. The coherence between GDP and LPI is a unique relationship and can answer several research questions. The abstract is appropriate and compact as it is. The title is acceptable as well; the authors try to follow it and more-or-less succeeded.

Comments of Authors:

Thank you for the comment.

Comments of Reviewers:

However, when it comes to literature review, the global approach can have disadvantages: the subparts are very short and not comprehensive enough. We see a quick run over the continents, however, a very draft and inappropriate analysis only. So, I recommend the extending of the literature where international sources are processed in a critical, analytical and comprehensive way.

Comments of Authors:

Thank you, your comment is well received. Considering the no. of words of the journal, we have revised the literature review and following 06 references have been added:

“Further, landlocked countries – Central Asian economies and a few South and Southeast Asian economies require to pass through another country(transit) to access major international transport lanes and connect with the global markets. Thus, these countries are reliant on the infrastructure availability of transit countries faces additional challenges in cross-border trade [20].

(Page 5, lines 10-14).

“Another study across ASEAN countries revealed that custom-related barriers such as time-consuming documentation requirements, burdensome inspection requirements & inefficient inbound clearance processes as well as mode-specific barriers such as Aviation cabotage regulations and limitations imposed on fleet size, equipment usage and hours of operation hinder the trade relations [22].

(Page 5, Line 16 - 21).

“Other studies have explored the influence of corruption on trade facilitation and revealed that corruption significantly affects the LPI [23].

(Page 5, Line 21-22).

“Saidi, Mani [28] too reveals that increases in transport infrastructure results in enhanced economic growth. Central and Eastern Europe countries possess strong economic relationships with countries in Western Europe resulting in increased employment and technology transfers. This can be further enhanced by developing infrastructure logistics which in turn strengthens connectivity across the regions. Other studies have shown that the LPI sub-components of logistics quality and competence as well as international shipment have positive significant impact on trade volumes [1]. 

(Page 6, Line 8-12).

“Other studies have shown that the LPI sub-components of logistics quality and competence as well as international shipment have positive significant impact on trade volumes [1].

(Page 6, Lines 13-14).

“Further, the logistical performance of the countries within this region has more impact on the export volumes in comparison the logistical performance of the importing countries [33]. In terms of the sub components of LPI, timeliness component is the most important for this region.

Page 6, Line 31 – page 7, Line 2).

“From the 6 sub-components of LPI, only two “competence of logistics services” and “quality of trade and transport-related infrastructure” have a weak, yet positive correlation with GDP per capita: indicator of economic growth [38]. In reality, the ability of Sub-Saharan Africa nations to link to global value chains is heavily influenced by the regional dimension of infrastructure and trade facilitation policies [39]. Thus, improving logistics performance in terms of infrastructure can have a positive impact on exports and trade facilitation across Africa.

(Page 7, Lines 22 – 28).

Martí, Puertas [33] has found that the within the South American continent the logistic performance of their own countries bears greater weight on the export volumes, as opposed to the logistics performance of the importing country. Further, LPI sub-components of international shipments as well as customs and tracking are highly significant with trade within the South American region. 

(Page 8, Line 11-15).

Investigating on the overall logistical performance, Dimitrievska, Mihajlović [24] has found that Oceania countries (together with Asian countries) come in second to the Europe region. Imports from Australia to Pacific Islands are heavily influenced by their population and per capita GDP [25]. The findings of the study imply that the population of Pacific Island countries and Australia, as well as the infrastructure of Pacific Island countries and their distance from Australia, have a substantial impact on their exports. 

(Page 5, Line 25 – 30).

---

## [Decision Letter · Decision Letter 1]

11 Feb 2022

Gross Domestic Product and Logistics Performance Index Drive the World Trade: A Study based on All Continents

PONE-D-21-35917R1

Dear Dr. Ruwan Jayathilaka,

We’re pleased to inform you that your manuscript has been judged scientifically suitable for publication and will be formally accepted for publication once it meets all outstanding technical requirements.

Kind regards,

Wen-Wei Sung, M.D., Ph.D.

Academic Editor

PLOS ONE

Reviewers' comments:

Reviewer's Responses to Questions

**Comments to the Author**

1. If the authors have adequately addressed your comments raised in a previous round of review and you feel that this manuscript is now acceptable for publication, you may indicate that here to bypass the “Comments to the Author” section, enter your conflict of interest statement in the “Confidential to Editor” section, and submit your "Accept" recommendation.

Reviewer #1: All comments have been addressed

2. Is the manuscript technically sound, and do the data support the conclusions?

Reviewer #1: (No Response)

3. Has the statistical analysis been performed appropriately and rigorously? 

Reviewer #1: Yes

4. Have the authors made all data underlying the findings in their manuscript fully available?

Reviewer #1: Yes

5. Is the manuscript presented in an intelligible fashion and written in standard English?

Reviewer #1: Yes

6. Review Comments to the Author

Reviewer #1: All my comments have been addressed by the authors. I think this paper can be accepted in current form.

7. PLOS authors have the option to publish the peer review history of their article (what does this mean?). If published, this will include your full peer review and any attached files.

Reviewer #1: No

---

## [Editor Report · Acceptance letter]

24 Feb 2022

PONE-D-21-35917R1 

Gross Domestic Product and Logistics Performance Index Drive the World Trade: A Study based on All Continents 

Dear Dr. Jayathilaka:

I'm pleased to inform you that your manuscript has been deemed suitable for publication in PLOS ONE. Congratulations! Your manuscript is now with our production department. 

Kind regards, 

on behalf of

Dr. Wen-Wei Sung 

Academic Editor

PLOS ONE